# Microfluidic and Micromachined/MEMS Devices for Separation, Discrimination and Detection of Airborne Particles for Pollution Monitoring

**DOI:** 10.3390/mi10070483

**Published:** 2019-07-18

**Authors:** Daniel Puiu Poenar

**Affiliations:** VALENS Centre for Bio Devices and Signal Analysis, School of Electrical & Electronic Engineering (EEE), Nanyang Technological University (NTU), 50 Nanyang Avenue, Singapore 639978, Singapore; epdpuiu@ntu.edu.sg

**Keywords:** particulate matter (PM), airborne inorganic particles, particle separation, particle detector/sensor, impactor filter (IF), virtual impactor (VI), cascade impactors, granulometry, air quality assessment, microfluidics

## Abstract

Most of the microfluidics-related literature describes devices handling liquids, with only a small part dealing with gas-based applications, and a much smaller number of papers are devoted to the separation and/or detection of airborne inorganic particles. This review is dedicated to this rather less known field which has become increasingly important in the last years due to the growing attention devoted to pollution monitoring and air quality assessment. After a brief introduction summarizing the main particulate matter (PM) classes and the need for their study, the paper reviews miniaturized devices and/or systems for separation, detection and quantitative assessment of PM concentration in air with portable and easy-to-use platforms. The PM separation methods are described first, followed by the key detection methods, namely optical (scattering) and electrical. The most important miniaturized reported realizations are analyzed, with special attention given to microfluidic and micromachined or micro-electro-mechanical systems (MEMS) chip-based implementations due to their inherent capability of being integrated in lab-on-chip (LOC) type of smart microsystems with increased functionalities that can be portable and are easy to use. The operating principles and (when available) key performance parameters of such devices are presented and compared, also highlighting their advantages and disadvantages. Finally, the most relevant conclusions are discussed in the last section.

## 1. Introduction

The U.S. Environmental Protection Agency (EPA) has identified six “criteria pollutants” of great importance due to their major impacts on both health and the environment, namely: particulate matter (PM), carbon monoxide (CO), nitrogen dioxide (NO_2_), sulfur dioxide (SO_2_), ozone (O_3_), and lead (Pb) [1]. Here we will focus only on the detection and characterization of inorganic PM in miniaturized and especially chip-based realizations as they allow drastic downscaling and implementation of complex ‘smart’ microsystems with multiple functionalities. 

Initially two distinct categories of PM were defined: PM_10_ and PM_2.5_. The common understanding of the definitions of these two categories is that the particles in the PM_10_ class have average diameters between 2.5 and 10 μm, while those in the PM_2.5_ class would have diameters of up to 2.5 μm. However, this is not 100% correct. In fact, PM_10_ is actually defined as “particulate matter which passes through a size-selective inlet with a 50% efficiency cut-off at 10 μm aerodynamic diameter” [2]. The World Health Organization (WHO), EPA and the European Union (EU) indicate different maximum permitted levels for each category [3]. The coarse inorganic particles of the former category are commonly found near roadways and dusty industries, while the “fine” particulates of the latter category are emitted or formed through chemical reactions, fuel combustion (e.g., by burning coal, wood, Diesel fuel), industrial processes, agriculture (plowing, field burning), and unpaved roads [1].

While most of the mass is usually in the fine particles with sizes between 100 nm and 2.5 µm, the largest number of particles is found in the very small sizes, less than 100 nm. As it can be deduced from the relationship linking particle volume with mass, these so-called ultrafine particles often contribute only a few percent to the mass but represent over 90% of the numbers. They can have an even more negative impact on human health than the typical larger particles in PM_2.5_ and PM_10_ classes, as they can penetrate into the pulmonary and cardiovascular systems and give rise to lasting conditions, such as increased predisposition to heart diseases or to premature births and affect fetal development. For this reason, a separate class, PM_1_, has been recently defined and used by certain agencies (although not recognized by US or EU standards), e.g., to measure the chronic pollution in New Delhi [4,5].

In general, airborne PM comprises a complex mixture of organic and inorganic components. While the first definitions mentioned above did not explicitly exclude the bioaerosols, the subsequent descriptions for the sources of different PM categories implicitly considered only inorganic pollutants. Indeed, although organic particles can have an important impact on human health (pollen, mold spores, cat and/or dog dander and dust mite particles, debris and feces are the most relevant allergens considered together with inorganic PM for indoor quality assessment), pollution monitoring focuses exclusively on the outdoor observation and quantification of inorganic PM.

Therefore, this paper reviews the most relevant solutions for separation, detection and quantitative assessment of inorganic PM concentration in air with miniaturized, portable and easy-to-use platforms, with special focus on chip-based implementations. We believe that such a paper is a needed timely and valuable contribution, due to several key reasons. First, because exposure to fine and ultrafine dust, among the other air polluting agents with deleterious effects of their own, has been identified as a clear threat for human health although the exact mechanisms of action still need to be clarified. It has been widely demonstrated that there is a correlation between high concentrations of PM in the atmosphere, mostly in urban environments, and the increase of some pathologies, particularly of pulmonary diseases [6,7,8,9] but an important negative impact was also noted for cardiovascular diseases, especially for the ultrafine particles [10]. With increasing public health awareness, standards and rules for exposure limits of PM have been formulated by different governments and organizations. In most of current standards, the quality of the air is documented by the particle mass concentration in size fractions up to 2.5 μm (PM_2.5_), or up to 10 μm (PM_10_), e.g., in the EU the 24 hrs limit value of PM_10_ is 50 μg/m^3^ and in US the 24 hr limit values of PM_10_ and PM_2.5_ are 150 μg/m^3^ and 35 μg/m^3^, respectively [2,11,12]. Therefore, not only it is well known that air quality analysis is vital for human health, but lately it has become increasingly important and even necessary in the context of increased pollution, global warming and for assessing the severity of haze caused by wildfires and ‒in South-East Asia‒ by clearing of plantations, forests and peatland by burning. 

Second, the bulk of microfluidics literature deals with handling liquids and much less attention has been devoted to handling of gases, particularly for PM measurement. 

Third, chip-based realizations are the most relevant for the implementation of highly compact, low cost and portable, yet performant, microsystems. The fact that on-chip implementations (realized, e.g., by micromachining) can be easily integrated together with signal conditioning and data processing circuits or other added elements, such as microfluidics and/or micro-electro-mechanical systems (MEMS) structures, could provide many more functionalities, resulting in microsystems much more performant than current portable PM analyzers. The general target is the development of future devices that could be more compact, more sensitive and more performant than current equipment, promising to revolutionize this area of activity just as the lab-on-a-chip (LOC) trend has begun to revolutionize biology and medicine.

Lastly, but not in the least, very few reviews have been published dedicated to this specific topic, the latest similar work being the review of Carminati et al. [13].

Any air quality analyzer needs to comprise two important elements: one which first performs the separation (or selection) of only the particles in the size range of interest, and a detector which can provide an output proportional to the mass concentration and/or the number of particles in the unit of volume. Consequently, the paper will first examine the various methods that can be employed for airborne particle separation, after which the following section will detail the sensors which can be used for the detection and characterization of inorganic PM.

## 2. Particle Separation Methods 

### 2.1. Traditional Impactor-Based Designs and Definitions of Their Performance Characteristics

A relatively convenient and often used method to separate particles only in the desired PM class (e.g., PM_2.5_ only) is by using an impactor filter (IF) or a virtual impactor (VI). IFs are typically used to gravimetrically separate the desired particles in air according to their mean aerodynamic diameters (MADs). The operating principle of IFs employs the particles’ inertia to collect only the particles of certain desired sizes onto an impaction plate. For this purpose, the air inflow has to be directed through a narrow throat in front of a large flat surface which forces the aerosol stream to rapidly and strongly change its flow direction laterally after accelerating. Only particles with small inertia can follow such a sharp and intense directional change of the air flow, whereas large and/or very dense particles are collected onto the impaction plate, which should preferably be covered with a very sticky layer (e.g., high-vacuum grease) to prevent the larger/denser particles bouncing off the impaction plate. The lighter particles continue their flow, typically towards a detector. A widely used geometrical design is the ‘Marple-type’, shown in Figure 1A [14], so called as the classical IF design was developed earlier by Marple et al. [15,16], with some subsequent improvements or modifications by other researchers [17,18,19,20].

The virtual impactor (VI) has the same operating principle as the IF, only that the coarse particles are not trapped onto a collection plate, but—as shown in Figure 1B [21]—form a central ‘minor’ particle stream (c) which is simply exhausted to the outlet, while one of the ‘major’ lateral jets (b) directs the separated particles towards the detector.

Because the single most important role of the VIs and IFs is separation of particles according to their mean aerodynamic diameters (MADs), their most essential performance characteristic is the so- called cut-off point (or, shortly, cut-point), representing the MAD value for which collection efficiency equals 50%. The collection efficiency (CEf) is defined as the ratio between the number (or mass) concentration *C_D_* of the collected PM (either onto the detector, or onto the collection plate, depending on which collection efficiency is desired to be calculated) and the total initial number (or mass) concentration *C_in_* of the PM introduced at the device’s inlet:CEf = *C_D_*/*C_in_*, (1a)
or, alternatively, a separation efficiency *η* for the particles of interest (i.e., those which will subsequently be detected by the particle sensor further in the device) can be defined [17]:(1b)$$\eta =\frac{C_{m}}{C_{M}+C_{m}},$$
where *C_M_* and *C_m_* are number (or mass) concentrations of classified aerosols in the major and minor flow paths normalized with respect to the input flow, respectively [17].

From Equation (1a) it results that, in the case of the VI, CEf can be redefined as the ratio of (light and/or small) particles that follow the ‘major’ flow streams to the total number of particles that enter the inlet, and which increases with decreasing particle size. Indiscriminate size-independent particle losses onto the walls represent a significant element that leads to the reduction of CEf, particularly in older, bulky macroscopic instruments. Figure 2 illustrates the variation of the key characteristic parameters with the dimensionless Stokes number *Stk* defined as [14,17]: (2)$$Stk=\frac{\rho CD_{p}^{2}V_{0}}{9\mu W}\;,$$
where *ρ* is the particle density, *C* is the so-called Cunningham slip correction factor for a particle of diameter *D_p_* (correcting Stokes law of viscous resistance for smaller particles), *V*_0_ is the mean fluid velocity at the nozzle throat, *μ* is the fluid viscosity, and *W* the inlet’s nozzle diameter (see Figure 1A). The Cunningham slip correction factor is given by [14]:
(3)$$C=1+\frac{2\lambda }{D_{p}}\left[1.165+0.483\exp\left(-0.997\frac{D_{p}}{2\lambda }\right)\right],$$
where λ is the mean free path of particles in air.

Other parameters can also be defined and play an important role in the device design, such as the inlet’s jet Reynolds number *Re_j_*, the stopping distance *Ψ* for a given particle and jet size (i.e., the inlet’s geometrical dimensions), the Stokes number at 50% collection efficiency *Stk*_50_ (important because the maximum wall loss occurs for the particle size approximately corresponding to *Stk*_50_, as it can also be seen in Figure 2), and the nominal volumetric inlet flow rate necessary for the correct operation, typically given in units of (L/min) or (mL/min).

Both wall loss and (for the IF) the possible subsequent particle bounce and re-entrainment in the air flow cannot be accurately predicted, hence making it difficult to estimate the value of *Stk*_50_. In any case, it is evident that both these factors deteriorate the separation efficiency curve (especially in the cut-off region) and, therefore, they should be reduced to a minimum in order to have good (i.e., abrupt) and predictable cut-off characteristics. However, while particle bounce can be more easily dealt with by coating the collection plate with a sticky layer, minimizing wall loss is definitely much more challenging and can be present for both the IF and the VI.

In the case of cascaded impactors consisting of two or more sequential stages, additional characteristics may be defined and used, such as the cross-sensitivity between impactor stages.

An illustration of a microfabricated VI is shown in Figure 1B [21]. The colored flow lines in Figure 1B represent simulated particle traces for particles with MADs of 1 μm (green) and 5 μm (red) that first enter the inlet (a) and are ultimately separated in the so-called ‘major’ (b) and ‘minor’ (c) flows that contain coarse and fine particles with MADs above and below the desired cut-point (2.5 μm in this case), respectively. Label (d) refers to the narrow neck where the flow line spacing is altered [21]. 

Other micromachined VIs were also reported. Kim et al. [22] fabricated such a micro-VI using bulk micromachining, with an injection nozzle 156 μm wide and a channel more than 200 μm deep whose design cut-off particle diameter was initially set as 10 μm. However, this was a passive hydrodynamic particle sorter for classifying biochemical components in liquid media, not for airborne particles. Hence, the device was tested by sorting aqueous solutions of 3 and 15 μm polystyrene microbeads, as well as with V653 biological cells (average diameter of 10 μm) suspended in solution. For particles with a diameter of 10 μm (for which the collection efficiency should be 50%) the measured collection efficiency was 68%. The authors attributed this discrepancy to the fact that the actual particle diameters were larger than 10 μm [22].

Luo et al. [23] realized a simple micromachined VI with convex channels patterned in plexiglas while the actual VI and microfluidic channels were patterned in 100 μm thick polydimethylsiloxane (PDMS) bonded to a glass coverslip. The VI was designed for a cut-off particle size of 2.5 μm, having an inlet width of 200 μm while the lateral bypass channels width was 500 μm. Practical tests (performed at an unindicated air flow rate, presumably 3.6 mL/min) with 2 μm-diameter polystyrene microspheres confirmed that the cut-point of their VI was close to 2 μm. Although it was initially mentioned that the system was intended for air pollution assessment, subsequent tests were performed using *Escherichia coli* (EPR300) bacteria to verify the system’s ability to separate microbes and showed that the VI’s cutpoint changed for sizes between 3 and 4 μm when the air flow rate was of 6 mL/min [23].

Another micromachined VI was intended to be used not for inorganic particles but for bioaerosol collection [24]. As was mentioned in Section 1, organic PM forms another important component which can significantly impact human health. Hence, bioaerosol sampling and separation is essential for bio-safety analytical work, particularly for biological applications and assessment of indoor cleanliness in medical environments, where it is vital to monitor both qualitatively and quantitatively the types of existing viruses and bacteria and their concentrations. However, the separation and characterization of bioaerosols are distinct topics from those pursued here, and which also typically require very different technical solutions than those enumerated in this paper. Therefore, we shall not delve into a detailed exploration of this rich topic which would clearly need a separate review. The micromachined VI for bioaerosol separation mentioned above as well as another reported realization for a similar purpose discussed in subsequent Section 2.2 were included in this review to illustrate that the same fundamental principles used for inorganic PM separation can be used equally well for bioaerosols, and also because they were tested with inorganic particles. Regarding the micromachined VI for bioaerosol collection, its nozzle width *W* and thickness/height *t* had values of 880 and 200 μm, respectively, with a *S*/*W* ratio value set at 1.5 (*S* is the jet-to-plate distance, see again Figure 1A) for an inlet air flow rate of 0.33 L/min in order to achieve the desired cut-point value of 1 μm [24]. The device was fabricated in SU-8, with top and bottom poly(methyl-methacrylate) (PMMA) plates for tubing and a PDMS gasket for tight sealing. First the system’s collection efficiency and wall loss were assessed with polystyrene latex (PSL) microbeads with sizes of 0.5, 0.8, 1, 1.5 and 2 μm. The measured cut-point was of 0.95 μm and the measured collection efficiencies for all particle sizes matched very well the calculated ones. The measured wall loss had a maximum value of 33.5% for an aerodynamic diameter of 1 μm and was caused mainly by the turbulence of the flow and eddies generated around the tip of the collection channels. Subsequent tests were carried out with bioaerosols obtained from suspensions of *Staphylococcus epidermidis* (ATCC 14990) with a geometric mean diameter of 1.08 μm and a maximal bioaerosol size of 1.2 μm. The measured data indicated an overall physical collection efficiency based on the number concentration of 73.8 ± 3%, and the collected bioaerosols were more monodisperse that those input in the system, with geometric mean diameter and geometric standard deviation of 1.23 and 1.28 μm, respectively. Furthermore, most of the bioaerosols collected at the output were viable [24].

In an interesting implementation the VI was fabricated in solid-phase photoresist, known as dry film photoresist (DFP), by using rapid prototyping for all the system’s three-dimensional (3D) microfluidics and other elements, as it was stated that this method provided reduced cost, short processing time and ease of handling [25,26]. The width and thickness/height of the VI’s inlet were 1000 and 400 μm, respectively, while the minor-to-total flow ratio was designed to be 10% at an air flow rate of 0.3 L/min in order to achieve a good particle collection efficiency. The entire VI was microfabricated using eight layers obtained by mechanically folding the DFP and had a total size of 20 mm × 15 mm with a height/thickness of 400 μm. The VI was employed in two different detection systems. The first had a PMMA channel chamber and used a PPD42NS commercial particle sensor to continuously count the particles after their separation. The system’s performance was compared with that of a detection station and the authors concluded that their system basically reflected the same trend in concentration of particles [23]. The second system used thermophoretic deposition on an interdigitated sensor, with which capacitive and impedimetric sensitivities of ‒56.8 pF/μg and 3.68 Ω/μg were obtained, respectively, when tested with carbon black particles (probably 2.5 μm in diameter) [26].

An attractive realization was the electrically tunable VI realized by Kim et al. [27,28], with which the cut-off diameter can be tuned by applying an electric potential across two Al electrodes integrated in the VI, as shown in Figure 3. Nominally, the VI was designed with a cut-off diameter of 1 μm but the addition of the electrodes enabled the classification of airborne nanoparticles. This was possible because, when an electric potential is applied, the electrical mobility μ_e_ of the small-inertia particles is larger than that of the large-inertia ones due to an almost quadratic inverse proportionality of μ_e_ with the nanoparticle size. Consequently, the resulting movement of the nanoparticles is now dictated not by the geometrical features of the VI but by the voltage applied across the electrodes. The system was fabricated in 200 μm thick SU-8, included a pair of 2 μm Al electrodes (insulated with a 3 μm-thick polyimide layer in a first version [27], and with 300 nm PECVD SiO_2_ in a second version [28]), and was packaged with PMMA plates and a PDMS gasket for tight sealing. The characterization of the unbiased VI was performed using dioctyl sebacate (DOS) particles ranging from 100 nm to 600 nm and carbon particles ranging from 600 nm to 10 μm. The collection efficiency and wall loss variation were measured as a function of the MAD of input particles without applying any electrical bias and the results matched very well the theoretically calculated values. The wall loss at the cut-off diameter was almost 25% and the maximum value was 33.5% [28]. The electrical tuning performance was investigated with solid monodisperse NaCl particles ranging in size from 35 nm (15 nm in the second version) to 70 nm and the outcoming size distribution was analyzed using a scanning mobility particle sizer (SMPS). The experimental results confirmed that the collection efficiency decreased inversely with the particle diameter and increased linearly with the applied voltage. Also, the collection efficiency at a fixed voltage inversely decreased as the particle diameter increased, so that very low values of around 10% (or much less for low applied voltages) were obtained for the largest (70 nm) nanoparticles [27,28]. The key advantage of such a tunable system over the traditional VI is that it provides flexibility in usage when different cut-off diameters have to be chosen dynamically. However, very high voltages had to be used: even for this microscale device D.C. potentials between 1 to 3 kV had to be applied to successfully tune the cut-off diameter of the captured particles from 35 to 70 nm in the first device version [27], or between 0.25 to 3 kV for varying the cut-off diameters between 15 to 35 nm in the second device version [28]. A similar device was also realized subsequently by the same team, but with 4 electrodes [29]. It included not only two tuning/particle accelerating electrodes but also two current sensing electrodes (one each for a minor and a major port, respectively). The system classified nanoparticles by electrically accelerating them to a sufficiently high velocity, while also electrically tuning the cut-point that was not determined by the VI’s geometry. Monodisperse NaCl solid particles of 50 nm were classified by applying a 1.1 kV voltage to the acceleration electrodes. The final current of the NaCl particles (on the order of 4 fA but amplified 4× by the acceleration potential) was detected on the sensing electrodes after condensing them [29].

Details about the electrostatic principle and miniaturized devices used for PM detection are given in Section 3.2.3.

### 2.2. Cascade Impactors

Cascade impactors consisting of several sequential stages have also been studied and the design of such macroscopic devices was established in the late 1970s [30]. Recently, such cascade impactors have also been miniaturized. One example is the 2-stage discrete mini-assembly realized by Maldonado-Garcia et al. [31]. Both stages were micromachined in silicon-on-insulator (SOI) wafers and also included MEMS resonators, one or more micro-nozzles and a micro-chamber. The MEMS resonator was a small actuator beam placed between two large plates where the particles to be sensed were collected. The beam served as both a piezoresistive sensor and a thermal actuator which excited thermally an in-plane extensional resonance mode. Each impactor stage was fabricated on a separate chip; the first chip was attached onto a printed circuit board (PCB) (which also contained the other necessary circuits) and wire-bonded to it. The cut-off MADs for the two impactor stages were 105 nm and 9 nm, respectively. The second stage chip-scale impactor was packaged in a transistor outliner (TO-8) header and wire-bonded to the header pins, and the TO-8 was then attached onto the PCB right above the first stage. Its machined cap was also connected to a pump which maintained an air flow of 0.3 L/min. The testing of the system was performed by exposing it for 30 min. at a time in three different environments: regular laboratory, air purifier output and inside the cleanroom. The sensitivities of the resonators of the two stages calculated from the measured data were 0.63 Hz/pg and 18.3 Hz/pg, respectively [31].

A more complicated fully micromachined structure comprised two parts [32,33]: a cascade VI with three sequential stages to separate particles according to their own inertia, and a flow rate distributor to maintain the flow rates at the desired values for each stage and eliminate the need for supplementary pumps. The structures of both the cascade VI and of the flow distributor are shown in Figure 4. The three stages of the VI were designed to provide cut-off diameters of 200 nm, 2 μm, and 6 μm, respectively, for an inlet flow rate of 0.36 L/min. Each stage of the VI was patterned in separate SU-8 layers that were vertically stacked onto one another on a PMMA substrate, with thicknesses for the first, second and third stages of 600, 400 and 200 μm, respectively. The flow rate distributor was also realized in a SU-8 layer on a separate underlying wafer. The entire system was assembled with a PMMA jig and a PDMS gasket, and then was tested using dioctyl sebacate (DOS) particles sized between 100 to 600 nm and carbon particles from 0.6 to 10 μm with an air flow rate of 0.36 L/min at the inlet. The collection efficiency was then determined experimentally for each minor channel and the measured data matched very well the calculated ones. The actual measured cut-off diameters were 135 nm, 1.9 μm, and 4.8 μm. However, particles larger than 4 μm sedimented onto the walls instead of being collected at the first stage [32,33].

A similar microscale cascade impactor intended for bioaerosol sampling and separation was also realized and tested by Kang et al. [34]. Just as for the micromachined VI for bioaerosol collection discussed previously [24], the microscale cascade impactor of Kang et al. is also included here because it was tested only with inorganic particles, and cascaded impactors are useful not only theoretically but are also employed for inorganic PM applications.

The traditional system with which bioaerosol sampling and separation is performed is the six-stage Anderson cascade impactor [35], and it has significant shortcomings: it is very large and bulky, requires a large flow rate and its usage also has obvious bio-safety and cleanliness concerns of its own in order to extract the captured biocomponents and then clean and re-use its elements. In contrast, a microfabricated realization is portable, much safer and requires a reduced air flow rate, in this case 0.5 L/min. The microscale cascade impactor of Kang et al. [34] comprised a rectangular-shaped jet nozzle and three impaction stages, with designed 50% cut-off particle sizes for each impaction stage of 1.06, 0.55, and 0.26 μm, respectively. The device was fabricated in PDMS using a SU-8 mold 300 μm thick, and its performance was evaluated using PSL microbeads with sizes between 0.2–2.5 μm, as well as ammonium sulfate particles sized between 10–700 nm. In order to minimize the bounce and re-entrainment effects, the impaction zone was coated by spin-coating with a silicon oil layer with a thickness larger than 0.5 μm. The measured 50% cut-off particle sizes for each stage were numerically calculated to be 1.06, 0.55, and 0.26 μm for an inlet flow rate of 0.5 L/min., and the measured values were found to be 1.19, 0.51, and 0.27 μm, respectively. The measured particle losses in the system were 5%–9% for particles sized between 0.2–2.5 μm and 9%–12% for the 0.05–0.2 μm particles. However, the authors stated that particle losses for particles 1 to 2.5 μm in diameter were higher than those reported by Marple in a previous large macroscopic realization [36]. Although tested only with inorganic particles, the authors concluded that such a microscale cascade impactor can be a very convenient personal bioaerosol detector which is not only ultra-compact, cost-efficient and easily mass produced, but can also be further developed for a cartridge-type module of increased safety and convenient use in a larger portable system [34].

### 2.3. Other Particle Separation Techniques

An elegant method to separate particles by size is based on aerodynamic effects, namely using the centrifugal force exerted on each particle when it is in an airstream flowing in a curved microchannel [37]. In one such realization, the designed microchannel was 60 μm wide and 100 μm high. After an initial region for hydrodynamic focusing of the particles, it had a 180^°^ bend with a radius of curvature of 2.5 mm, followed by an 18 mm long straight section after the bend, at the end of which the channel gradually widened to a final width of 2 mm over a length of 6 mm (see Figure 5a). Numerical simulations indicated that this design would provide a clear separation between particles with sizes of 0.2 and 0.75 μm travelling at a 1.5 m/s average air velocity. The separation efficiency dependence on the flow rate was also calculated for a variety of particles with differences in diameter of either 0.5 or 1 μm. The calculation results revealed that each size range of particles had its own optimal velocity for separation. Beyond this optimum, the larger particles will reach the outer wall in the circular portion of the structure at much smaller velocities than the smaller ones and an overlap in the particle distributions will occur both at the top and bottom of the channel, limiting the achievable separation resolution. The practical device was fabricated in PDMS from a SU-8 mold and was tested at three different air velocities using fluorescent-dyed PSL microspheres with diameters in the range 0.2 to 1.9 μm. The measured results showed that more than 80% of the large sized particles were significantly separated into the middle and radial outer channels, respectively. The smallest of the particles, 0.2 μm in diameter, were nearly equally split between the radial inner and middle outlets. Increasing the air velocity would shift the highest separation towards the smallest particles, but ‒at the same time‒ this also decreased the largest particle size that could be transported through the channel. The system has a simple structure that does not require a complicated fabrication process and is easy to use, without the need for any additional power supply (like electrostatic separators). However, it also has a few drawbacks due to several practical factors. The first is the inherent low velocity near the side walls characteristic for a typical parabolic flow profile, which would prevent a proper separation of the particles located at extreme positions closest to the walls. Hence, performing hydrodynamic focusing of the particles before performing the separation becomes absolutely necessary, thus complicating the design and the necessary overall set-up. Second, if it is desired to separate the particles into a larger number of categories (‘bins”), the fluidic design of the structure can become extremely complicated. Finally, in practice the airborne particles may have aspherical shapes which would also impair the separation precision and efficiency. Nevertheless, if only a simple separation into one or two categories is desired (e.g., PM_2.5_ and/or PM_10_), this method is indeed a very attractive alternative [37].

A very different separation approach utilized an array of I-shaped pillars inside a microfluidic chip realized by Yin et al. The pillars were arranged in slanted lines and with positions in a line shifted with respect to those in the line above, as illustrated in Figure 5b [38]. However, the pillars were not used to perform actual filtering of the particles. Instead, the pillar array induced a deterministic lateral displacement (DLD) of the particles in the fluid. The DLD, first investigated by Huang et al. [39], makes use of the asymmetric bifurcations of a laminar flow around obstacles ‒in this case the I-shaped pillars‒ that cause gradual deflection of larger particles, resulting in final size-based particle separation. Despite the more random nature of the localized particle-pillar interactions on the overall the paths of the particles are ultimately dictated deterministically by the individual size of the particles. As a result, Yin et al. stated that a high separation efficiency with very high size resolution could be obtained in their chip with the I-shaped pillars array, although no numerical values were provided in support of this statement [38]. The DLD principle had been applied earlier to liquid samples, e.g., to successfully separate nonspherical cells (red blood cells) with 100% efficiency from a blood sample diluted ten times in phosphate-buffered saline (PBS) buffer [40]. By carefully designing the pillar size, spacing and array gradient Yin et al. realized a small DLD microfluidic chip (with the main fluidic chamber 2 mm wide, 20 mm long and 15 μm deep) that could separate particles smaller or larger than 2.5 μm, respectively, at two different outlets [38]. Subsequently, electrochemical sensors were used for particle detection in order to achieve high sensitivity and simplify the instrumentation compared to conventional optical methods. In this case a three-electrode AC electrochemical impedance system detected the particles after separation. However, although the system was supposedly intended for PM_2.5_ monitoring of airborne particles, it was not tested with airborne particles but using liquids. Specifically, distinct solutions of polystyrene beads with 1 μm and 10 μm diameter were employed (unmixed, i.e., each used separately), with the latter stated to have been separated with an efficiency of nearly 100%, but no other performance characteristics were given. Moreover, for the electrochemical detection method to be fully usable, a dilute electrolyte solution was also added to each particle-containing solution [38].

Another fluidic-based separation method, used by Kuo et al. [41], employed microcentrifuge tubes of small volume (e.g., 0.2 ml) as a cyclone separator, a solution which was stated to effectively decrease the size and power consumption of the final device. The cyclone geometry consisted of a cylinder vertically stacked and joined onto an inverted truncated cone. The particles with MAD values larger than the desired cut-off value (2.5 μm in the case tested by the authors, with 0.5 mm diameters for both the inlet and outlet of the cyclone separator) were collected at the bottom of the tube, whereas particles with MAD values smaller than the cut-off value were transported to the upper outlet [41].

## 3. Particle Detectors and Their Performance Characteristics

In order to detect the presence, number/concentration, size and/or total mass of the particles of interest in the airstream various types of detection principles can be used, such as scattering or electrical, and they will be examined in the following sections.

The performance characteristics of the particle detectors can be expressed by the well-known parameters used to quantify the performance of any detector, such as limit of detection (LOD), sensitivity, linearity (of the detector’s output vs. input transfer characteristic), accuracy (precision), resolution, etc. The units used for each parameter will, of course, need to correspond to the desired parameter to be measured (number or mass concentration, or even total mass of particles) as well as the key parameter describing the operation principle of the detector. Here we shall use the clear definitions for performance characteristics of any detectors/sensors formulated by Fraden [42].

For instance, let’s assume that a micromachined resonator is used to detect the mass concentration of particles attaching onto it, particles that had been separated earlier in the device. As it will be also detailed later, the operation principle of a resonator is to provide an oscillatory electric signal of a given resonance frequency (on the order of, e.g., hundreds of kHz or MHz, or even higher, depending on the type and size of the resonator), whose value will shift due to the accumulation of particles onto the detector’s surface. Then, for this type of detector, the LOD will be expressed in units of “mg/cm^3^” or ‒much more probable for an on-chip realization‒ in “μg/cm^3^” or even “ng/cm^3^”. The sensitivity in this case will be defined as the ratio between the resulting resonance frequency shift Δ*f* and the input variation of mass concentration of particles Δm that caused it. Hence, the sensitivity (*S* = Δ*f*/Δm) will be expressed in units of “Hz/(μg·cm^‒3^)”. Its precision/accuracy, defined as the difference between the highest deviation between the detected value and the ideal value (i.e., the actual, real value, measured with a highly accurate and very well calibrated reference measurement tool) [42], would then be given in the same unit as the measured parameter, e.g., (in this example) in “μg/cm^3^”. However, sometimes accuracy can be defined like the relative error, i.e., as the previously mentioned difference between the detected and the ideal values divided by the ideal value [42], in which case it should be given in “%”. Lastly, the detector’s resolution, defined as the smallest increment in the measurand that results in a clear output change [42], would then also be expressed in “μg/cm^3^”.

Unfortunately, only some of the performance characteristics are usually provided in papers found in literature, most probably because of the limited budgets and time available for the researchers, as well as due to the difficulties necessary to be overcome in order to measure accurately and reliably all the performance characteristics with a single setup. Furthermore, even most of the commercially available handheld devices do not provide all these performance characteristics. This is evident from Table A1 in the Appendix A, which compares the key performance parameters of commercially available handheld/miniature particle sensors. In almost all cases just some parameters are specified, but even the definitions and measurement units used for a few of the performance specifications are not always the standard ones explained above, which only increases the difficulty for the users in making direct comparisons between devices.

Likewise, just as for the separation devices presented earlier, there will be a very large variability in the performance description of the particle sensors reviewed in the following sections. This is due to both substantial differences in measurement setups and methods, and significant variations in the amount of information provided, with a limited amount of data given in short Conference papers. This highlights another key role of review papers like this one, namely to extract and concentrate the essential information in a concise and compact format for the readers to be able to have a better understanding of the main characteristics of previously reported devices of the same type.

### 3.1. Optical Detectors

Light scattering is the most often used particle detection principle and is almost universally used in handheld and/or portable air quality analyzers that are currently commercially available. Hence, it is not surprising that many reported miniaturized or on-chip realizations have employed light scattering as well. 

A complex system was realized by Schrobenhauser et al. [14,43], though not in an on-chip realization but in a miniaturized system that first separated the particle of interest also by using inertial filtering with an IF, as presented earlier. The particles then crossed a laser beam and the light was collected and focused by a system of three Fresnel lenses, as it can be seen in Figure 6a [14,43]. The first, frontal Fresnel lens spanned around a laser beam trap at the center and it collected and collimated the scattered light. The other two Fresnel lenses, located behind the frontal one, were off-center excentric ring lenses that focused on two avalanche photodiodes (APDs) with which the intensity ratio of scattered light at two known solid angles was measured, from which the particle size could be deduced. Subsequent additional statistical analysis and signal processing were used to further enhance the signal-to-noise ratio (SNR) of the signals detected by the APDs and the general accuracy of the measurement method. The measured results were in good agreement with theoretical expectations when tested with polystyrene latex (PSL) and silica particles in a size range of 300–1600 nm (though in practice the sensor proved capable of measuring particles in a size range from about 140 nm up to several microns), with a particle size determination accuracy of at least ∼10–20% for monodisperse particle solutions [14,43]. 

Another miniaturized implementation also employing discrete devices is that of Yuen et al. [44]. Their mini-system comprised fluidics with a particle trap impactor and a relatively complex light scattering detector, all packaged in a 3D-printed enclosure. The particle trap impactor consisted of a T-shaped fluidic junction fabricated in PDMS, with inlet on the left, the large particles outflowing through the channel at the right, and the separated particles with sizes in the relevant range of 0.5 to 2.5 μm exiting the outlet at the bottom and being directed into the scattering detector. This was a forward light scattering detector comprising a laser diode, two lenses, a light trap patterned on the collection lens, and a photodiode, with the particle flow path normal onto the light propagation direction, placed between the focusing lenses of the laser diode and of the photodiode. The particle trap impactor’s performance was characterized at 0.5 L/min. inlet flow rate with particles sized from 0.5 μm to 3 μm in 0.5 μm increments and the measured separation efficiency matched the simulated values very well. The scattering detector was tested with 1.5 μm particles and the particle count it provided was directly compared to that provided by a reference aerodynamic particle sizer (APS). The measured particle counts exhibited good linear correlation with those of the APS in the range 30–250 counts, with a counting efficiency estimated to be ~0.5% lower than that of the APS [44].

A simpler monolithic scattering-based PM_2.5_ detection system realized on-chip by Li et al. [45] was formed by two stacked silicon wafers which were joined using adhesive bonding. The bottom wafer hosted a square measurement chamber with a 2.5 mm side as well as the interconnections. The top wafer only contained the microfluidic channels for the air flow and the inlet and outlet through-holes. The chamber was fabricated by anisotropic wet etching of the Si substrate to a depth of 300 μm, and the Al electrodes were deposited and patterned not only on the wafer surface but also in the chamber and on its slopes. A laser diode (λ = 650 nm) was used as the light source, mounted in a recess in one of the corners of the chamber. A photodiode was placed in the diagonally opposite corner of the chamber and orientated normally onto the laser diode’s emission direction. The air inlet and air outlet were at the other two corners. The schematic structure is shown in Figure 6b. The separation of the particles with sizes of interest was achieved by using a downward pointing size-selective inlet that balanced particle size-dependent gravitational settling velocity against the upward sampling velocity [45]. Improved subsequent versions of this sensor included an integrated VI that was also realized in the top wafer and exhibited a limit of detection (LOD) of ~10 μg/m^3^. The mean absolute errors and standard deviations were calculated between the measurements performed with a GRIMM aerosol monitor (GRIMM Aerosol Technik Ainring GmbH & Co. KG, Ainring, Germany) used a reference and those obtained with the tested sensors. The measured results indicated good capability of fast and reliable detection of indoor airborne particles, with an accuracy (for the latest, most advanced version of the device) as high as 2.55 μg/m^3^ [45,46,47].

### 3.2. Electrical Detectors

#### 3.2.1. Resonant Sensors

##### Quartz Crystal Microbalance (QCM) and Surface Acoustic Waves (SAW) Sensors

The sensors in this category provide an electrical output but the basic sensing operation principle is not entirely electric as it relies on the electrically-induced generation and propagation of ultrasonic waves in the bulk, or at the surface, of a piezoelectric material. The quartz crystal microbalance (QCM) is a piezoelectric plate with two metal deposited electrodes, one on the top and one on the bottom face. The QCM works as a bulk resonator system, having a fundamental oscillation frequency correlated to the physical system properties that can change following a mass deposition on its electrodes, like that resulting from accumulation of PM. 

High frequency SAW devices consist of interdigitated transducers (IDT) and reflectors deposited onto a piezoelectric substrate (e.g., LiTaO_3_) using metal thin films patterned symmetrically on each side of the device in order to produce a standing acoustic wave (e.g., Love or Rayleigh) along the surface of the substrate. The structure of such a SAW-based PM detector is shown in Figure 7. The mass sensing region, also called the active area, lies between the two IDTs patterned on the top surface of the substrate at some distance from one another. The reflector gratings patterned behind both IDTs form a resonant acoustic cavity which will trap the acoustic wave at the resonant frequency. As a result, standing acoustic waves will form in the active area, whose propagation is affected in terms of both velocity and amplitude when additional mass (which can be due to just a few molecules in the case of gas sensors) is present on the sensor surface. Therefore, the accumulated mass can be measured with high accuracy as the resonant frequency shifts if the sensor is used in an oscillator. However, the specific sensing mechanism of a SAW sensor may also depend on the mechanical and electrical properties of the captured particles [48,49,50]. Both QCM and SAW devices need to use a piezoelectric material as the substrate, with monocrystalline quartz as the traditional material of choice for QCM due to its excellent thermal stability, availability as wafers for convenient fabrication using standard microelectronic processes, and the capability to select the vibrational mode (thickness shear, width extensional, length/width flexure, etc.), the operational frequency range and other characteristics (noise, aging/stability) by choosing one of several possible cuts that have defined angles to the crystallographic axes, denominated as AT, BT, SC, etc. However, nowadays other materials are widely used, such as barium titanate (BaTiO_3_) and lead zirconate titanate PbZr_x_Ti_1−x_O_3_ (0≤x≤1) compositions, which are now the most commonly employed ceramic for discrete transducers. These materials are now predominantly used instead of quartz not only because they have more advantageous costs, but also because of their different piezoelectric properties, which ‒for the latter PbZr_x_Ti_1−x_O_3_ family‒ can also be changed by adjusting the composition of the material, and can also exhibit other useful sensing properties, e.g., pyroelectricity. Moreover, SAW sensors can also be realized as thin-film bulk acoustic resonators (FBARs or TFBARs) using the previous materials or others, such as aluminum nitride (AlN) and zinc oxide (ZnO), which can be easily coated on top of other devices or materials, such as Si wafers, thus offering better capability of integration in LOC smart systems [51,52,53,54,55,56] and/or with signal processing integrated circuits (ICs).While a great number of both QCM & SAW realizations have been reported, here just a few relevant illustrative examples will be given.

A QCM-based particulate measurement system (QPMS) was recently reported [57]. It consisted of four main parts: a fluidic sampler that fed the sensor surface only with particles having MADs larger than 0.8 μm at an inlet flow rate of 0.8 L/min., a particle mass concentration sensor comprising a grease-coated QCM with a microheater and a resistive temperature detector (RTD) integrated on the QCM’s quartz plate surface, a mini-pump and an electronic control board. By adjusting the working temperature, the PM collection onto the substrate could be changed. Thus, the sensor’s frequency shift observed at 80^o^C was about 2.5× greater than that at 20 ^o^C due to a different coating collection capacity when subjected to the same PM dose. Measurements indicated that, at 80^o^C, the system had a resolution of ~15 μg/m^3^ for a measurement time of 60 s (i.e., 1.5 μg/m^3^ for a measurement time of 10 min.), and a resulting calculated sensitivity of about 0.2 Hz/(μg/m^3^), or, when normalized to the device’s active area, of 0.095 Hz/(ng^.^cm^‒2^) for a 60 s measurement [57].

However, Kuo et al. [42] stated that SAW devices operating in a shear horizontal mode (SHM, i.e., a Love wave) can be more accurate and more sensitive PM_2.5_ sensors than QCM ones, while at the same time having a simpler fabrication process and requiring a simpler measurement setup. They built and tested such a SHM-SAW device as a PM_2.5_ mass concentration sensor. The structure also comprised a small (volume of 0.2 ml) microcentrifuge-based cyclone separator that enabled the separation and subsequent detection of particles with aerodynamic particles equal or smaller than 2.5 μm. Their SHM-SAW sensor had a LOD of 11 μg/m^3^ and a sensitivity of 9 Hz/ng, with good linearity up to a mass concentration of 107 μg/m^3^. It was stated that the error of the measurement was much smaller (but unspecified quantitatively) when the SHM-SAW sensor was operated together with the cyclone separator, but the sampling time had to be increased to 160 s to allow sufficient particles to deposit onto the sensor for detection [42]. Another realization also employed Love wave SAW sensors on an AT-cut quartz substrate in conjunction with a cascaded impactor that separated the desired particles (either PM_2.5_ or PM_10_) before they reached the sensor’s surface. The authors stated that a high sensitivity of up to 0.2 Hz per μg/m^3^ could be achieved for PM_2.5_ and PM_10_ particles in the 0–400 μg/m^3^ concentration range [58]. 

Thomas et al. [48] employed Rayleigh SAW sensors operating at 262 MHz, fabricated on 4-inch ST-cut quartz wafers with either free or electrically shorted active regions. The latter were obtained by covering the sensing area with a thin conducting layer such as gold, resulting in a sensor response dictated only by the mechanical properties of the captured particles. As expected, ’free’ SAW sensors had greater mass sensitivities than shorted ones as they exhibited larger frequency shifts. The system did not include any dedicated separator that would filter out the particles with sizes outside the range(s) of interest, and testing was performed by gently rubbing a bondwire onto the sensor surface using a XYZ microtranslation stage, resulting in deposition of Au particles with diameters <1 μm onto the active region. However, the sensing principle exhibits an intrinsic dependence of its sensitivity on the particle size because full acoustic coupling to the entire particle volume occurs only when the particle size is smaller than the acoustic penetration depth (∼3 μm for their device). Consequently, the sensitivity was much higher—in the range of Hz/ng—for particles having sizes comparable or smaller than the penetration depth, compared to values in the range of Hz/μg for particles larger than the penetration depth. Experiments carried out by depositing different types of particles with varying sizes confirmed this hypothesis. Such a 262 MHz SAW sensor detected masses below 1 ng with a high mass sensitivity of 275 Hz/ng [48]. The same group also reported a hybrid microsystem that included a virtual impactor, two 2.96 μm thick ZnO resonators (one as reference, one as the actual PM_2.5_ sensor) fabricated on a 4-inch Si wafer and operating at a much higher frequency (894 MHz) than the previously mentioned SAW sensors, and an oscillator and an application specific integrated circuit (ASIC) chip that were realized using a 0.35 μm complementary metal oxide semiconductor (CMOS) process. All these components were mounted on a PCB and the entire system encased in a suitable package. Tests indicated a sensitivity of 7.5 kHz per μg/m^3^, reportedly superior to that of a commercial QCM device used for comparison [59].

Chiriacò et al. [60] realized a particles sensor that employed both Rayleigh SAW delay lines and electrochemical impedance spectroscopy (EIS) for particle detection, each detector comprising an array of eight sensing areas. The devices were fabricated on ST-cut quartz wafers on which Al IDTs for the SAW delay lines and Au IDTs for EIS were patterned, after which microfluidics were fabricated by replica molding with PDMS using a 200 μm high SU-8 hard master. Operating at 206 MHz, the SAW-based sensor was tested with polystyrene beads with diameters of 1 μm, 200 nm and 40 nm, respectively, and exhibited a sensitivity of 0.4 ^°^/ng and a LOD of 1.9 ng. However, the devices were not tested using airborne particles. Instead, the devices were assembled **after** the deposition of particles from a liquid suspension in a solution of potassium hexacyanoferrate (II/III). After deposition the suspension drop had been subsequently evaporated, then a second drop was deposited and evaporated, etc. The procedure was repeated seven times for the SAW sensors and 5 times for EIS sensors, respectively, in order to achieve increasing concentration of particles. Thus, the devices did not include any component for size differentiation of the particles and testing also did not consider characterizing the sensitivity for particles of different sizes or detecting the particle numbers/concentration; instead it focused on detecting the total mass only [60].

A 3D-printed VI with in-plane channels and a size of 24 mm × 24 mm ×10 mm was reported by Zhao et al. [61,62], who first used it in a miniature PM_2.5_ detection system that assembled the VI with a QCM-based particle sensor realized in an AT-cut quartz crystal. A thin photoresist film was spin-coated on the Au electrodes of the QCM sensor to improve surface adhesion properties and thus enable capturing the particles to be detected. The system was reported to successfully separate by size SiO_2_ powders with particle diameters in the range 0.5‒8 μm, and it completed the measurement of a PM concentration of 35 μg/m^3^ in only 23 seconds at an inlet flow rate of 0.27 L/min., with a theoretical mass sensitivity of 288 Hz/μg [61,62]. A similar VI was then used in a system realized subsequently by the same group, but modified in order to embed the QCM sensor directly into the VI [63,64]. The minisystem was then tested at an inlet flow rate of 0.5 L/min. with SiO_2_ particles with diameters ranging from 0.5 to 8 μm [64] (up to 10 μm in [63]) and its theoretical mass sensitivity was calculated to be ‒2.742×10^2^ Hz/μg. For comparison, the measured data indicated a good mass concentration sensitivity of 0.0554 Hz/min per μg/m^3^, with linear fitting time-dependent sensitivities extracted from measurement data of about 3.467‒3.7 and 5.01‒5.2 Hz/min at low and high concentration levels, respectively (the latter 1.5× larger than the low concentration). The mass resolution of the QCM sensor was calculated to be 3.47 ng, and the detected mass concentrations of PM_2.5_ particles in the two cases were 52.33‒57.2 μg/m^3^ for a measurement time of 40 min. and 75.5‒79.7 μg/m^3^ for a measurement time of 50 min., respectively, showing only ~7% deviations from the theoretical values [63,64].

Another miniature PM_2.5_ detector was realized comprising a VI for particle separation, a thermophoretic precipitator (TP) for particle capture and a SAW sensor for mass detection [65,66]. The TP used two Peltier elements on opposite sides of an air microchannel 200 μm deep in order to create a temperature gradient across it that removed particles from the airstream by thermophoresis and deposited them onto the bottom cold floor. The SAW device was realized on an ST-X cut quartz substrate, while the microfluidics were etched using inductively coupled plasma deep reactive ion etching (ICP-DRIE) in 2 silicon wafers that formed the upper and lower shells of the final structure. The SAW sensor and the Peltier elements were placed inside one arm of the VI impactor and in holes of the upper and lower shells, respectively, which were then bonded together. Tests performed only with monodispersed PSL particles 2 μm in diameter indicated a linear response, with a sensitivity of 93.96 (Hz/min)/(μg/m^3^), but the sensor’s responsivity decreased in time due to the increased accumulation of mass on its surface [65,66].

A similar MEMS microsystem for PM_2.5_ measurement also comprised a VI for particle separation, a TP for particle capture and a FBAR as the particle detector, plus the necessary electronic circuits [22]. The final system was made up of 3 wafers: a bottom quartz wafer contained the FBAR and CMOS dies mounted on it *a priori*, the middle Si wafer comprised the microfluidics with inlet/outlet, and a top fused quartz wafer that served as a transparent cap for the air-microfluidic channels, as well as substrate for the thermophoretic heaters. They were all bonded using epoxy and then cured on a hotplate overnight. The experimental results have shown that the prototype had a maximum theoretical detection limit of approximately 2 μg/m^3^ with a 10 min. integration time [22].

From the examples presented here, it becomes clear that, although QCM- and SAW-based PM detectors can be and have been miniaturized, it is extremely difficult ‒even impossible for QCMs‒ to use them in a fully single-chip smart microsystem in which all components are realized in, or onto, the very same substrate. This is mainly due to the critical requirement of employing a piezoelectric substrate, such as quartz, into which it is not possible to realize the other necessary elements of the system, such as microfluidics, sensors and signal processing circuits. SAW, and especially FBAR, resonators can be realized in films deposited onto a substrate and, therefore, are more amenable to integration in on-chip microsystems. However, their usage/addition may complicate the fabrication process and/or the structure of the final system. In contrast, the MEMS or nano-electro-mechanical system (NEMS)-based resonators allow a much easier integration as well as significantly improved performance due to their much higher sensitivities. The next sections will present such realizations which were implemented in silicon and, therefore, can be used to realized fully integrated on-chip smart measurement microsystems.

##### PM Detection Using Micromechanical Elements (Cantilevers, Membranes, etc.)

MEMS elements (e.g., cantilevers, membranes, etc.) fabricated by micromachining are yet another category of devices whose electromechanical resonance is also very sensitive to mass variations. Unlike QCM and SAW devices, MEMS resonators are more attractive because they do not mandatorily need a piezoelectric material and can be fabricated in cheaper silicon wafers, although silicon’s piezoresistivity can sometimes be used instead for the operation of the resonator. The key advantage of MEMS/NEMS-based resonators compared with the conventional SAW and QCM ones is that they can provide orders of magnitude higher mass sensitivity and resolution due to their much reduced sizes.

One first example has already been mentioned earlier, in Section 2.2, when a cascade impactor using MEMS thermal piezoresistive resonators was presented [32]. Another example is that of a microcantilever which was electrothermally excited to achieve in-plane resonance at 218.59 kHz [67,68]. The rectangular cantilever had dimensions of 1000 μm × 170 μm × 13.5 μm and was heated up at 295.66 K using built-in resistors placed close to the cantilever clamp end with only a few mW power consumption. However, the practically achieved resonance was around 202 kHz because the practical thickness of the cantilever fabricated by bulk micromachining was 18.5 μm instead of the theoretical design value of 13.5 μm. A piezoresistive bridge was also implemented on-chip to monitor the response to in-plane vibration. In order to use the cantilever as a particle sensor, arrays of vertical Si nanowires (NWs) were selectively defined onto the surface of the micromechanical cantilever. Practical measurements performed using cigarette smoke aerosol indicated a reduction of the resonant frequency due to the deposition of particulate matter. The frequency shift was 1.5× larger when NWs were added, confirming that they roughened the cantilever surfaces and increased the amount of captured PM [67]. This particle sensor was then employed in a cantilever-based airborne nanoparticle (NP) detector system named CANTOR-1. For this purpose, it was mounted in a cylindrical tube with an inward-oriented and tilted conical inlet that directed the aerosol flow (at 0.68 L/min.) against the cantilever. At the same time, the cantilever ‒now operated at 361 kHz‒ was biased using high negative voltage (while the surrounding tube was grounded) to form an electrostatic sampler that would attract charged or polarized NPs towards it. Tests performed with carbon and TiO_2_ NPs indicated a possible NP mass of 203.09 ng detectable within 15 min., a mass sensitivity of 5.95 Hz/ng and a sampling efficiency of 1.33% [68]. Although this is an extremely small value, very large arrays of MEMS resonators that can cover a much larger surface can easily be realized, and this can enhance dramatically the overall efficiency. The sampler head, measurement chamber and/or the flow profile of a system like CANTOR-1 can be modified and optimized to provide maximal interaction of the cantilever(s) with airborne particles. Additionally, other trapping mechanisms may also be employed to further boost the particle trapping efficiency, e.g., electrostatic trapping, which was indeed used in the subsequent version CANTOR-2 detailed below.

Subsequently, an improved handheld version CANTOR-2 was realized by combining the cantilever sensor together with a miniaturized electrophoretic aerosol sampler, a microfilter that retained particles larger than 2.5 μm, an IF that eliminated particles larger than 1 μm from the airflow, a PCB and a high voltage supply module [69,70]. The new improved version utilized a square inlet chamber instead of the previous cylindrical one, and also reduced the input air flow to 0.33 L/min. but the cantilever was still biased at a high negative voltage (‒500 V). Practical tests were carried out by nebulizing a suspension of carbon NPs in a solution of water/ethanol or water/isobutanol using a constant output atomizer followed by a water trap, which finally resulted in the generation of two size modes of NPs with diameters of ~20 nm and ~120 nm, respectively, at total number concentrations in the order of 104 particles/cm^3^. Measurement results indicated a very good correlation of the CANTOR-2 data with the fast mobility particle sizer (FMPS) reference as the data measured by the former deviated from the latter with only 8–14% (~10% on average) and a limit of detection of ~5 μg/m^3^ after an integration time of ~3 min. was obtained, with a response time of 6 s [69]. Other tests, carried out with the cantilever now biased at +500 V and using TiO_2_ and SiO_2_ aerosols with diameters of 250 and 330 nm, respectively, as well as with cigarette smoke particles, resulted in a resolution of ~10 μg/m^3^ and also indicated that a high correlation (R^2^ > 0.95, up to 500 μg/cm^3^) was obtained with respect to the FMPS reference measurements. It was also found that the sampling efficiency of the equipment’s electrostatic precipitator scaled with the inverse of the NP diameter [70].

The same detection principle was applied at even smaller scales, using silicon nanopillars with diameters of e.g., 650 nm and aspect ratios of more than 60 [71]. These nanostructures exhibited a bending resonant frequency of ~484 kHz for the fundamental flexural mode. Practical tests were performed by nebulizing a suspension of TiO_2_ NPs in water/ethanol followed by a dryer. However, the vibration was not generated in the nanopillar chip itself but obtained by mounting the chip with nanopillars onto a piezo shear actuator and a nanometer aerosol sampler was used as an electrostatic precipitator (ESP) to collect the flowing TiO_2_ NPs. Experimental tests of the ESP operated at ‒9.5 kV under a constant particle-laden air flow of 1 L/min indicated that the NPs collected after a sampling time of 30 min. with an efficiency of ~3.5% had diameters of ∼125 nm with number and mass concentrations of 2.87 × 10^7^ particles and 121.53 ng, respectively. From the corresponding resonant frequency shifts a mass resolution of ∼1.5 fg (femtograms = 10^‒15^ g) and a mass sensitivity of 7.22 ± 0.14 Hz/fg were then deduced, which are orders of magnitude better than those typically achieved with SAW, QCM or FBAR technologies. Furthermore, and extremely importantly, two methods of NP removal were successfully demonstrated in order to make the nanopillar sensor reusable. The first one employed casting polydimethylsiloxane (PDMS) onto the pillar surfaces polluted with the previously captured NPs, which were now trapped in PDMS after curing and could thus be easily removed together with the polymer. However, the method could not be employed for thinner (d< 800 nm) and longer (h > 30 μm) pillars due to their fragility. The second method employed ultrasonic cleaning. For this purpose, the polluted nanopillar sample was placed in a glass with acetone solution or deionized water (DI) water, and the glass was then put into an ultrasonic bath containing DI water. Exposure to ultrasounds at 35 kHz for 2 min. cleaned the nanopillar surface with efficiencies of ∼97–99% [71].

A very different approach was developed in order to collect airborne NPs on a nanomechanical resonator with a high collection efficiency which enabled even the detection of single NPs [72]. In this case the nanomechanical resonator, a doubly-clamped beam, acted as a single filter-fiber. The airflow was directed through the sensor chip so as to pass by this filter-fiber and the airborne particles were collected onto it either by Brownian diffusion or inertial impaction, depending on their momentum. Hence, at low velocities diffusion is dominant and very small particles can be trapped, while at large velocities the NPs are carried away with the air flow and only large particles are trapped on the fiber by inertial impaction. The nanoresonators, a few microns wide and more than 135 μm long, were fabricated from 100 nm or 220 nm silicon nitride covered with a 50 nm thick Al layer on the front-side. They were excited magnetically by placing them perpendicularly to a static magnetic field and injecting in them an alternating current. Silica and silver aerosols were generated by nebulizing an aqueous colloidal sucrose solution resulting in nanoparticles with sizes of 25 and 100 nm, respectively. The mass increase induced by the homogeneous nanoparticle deposition on the resonant filter-fibers with a collection efficiency of up to 65±31% produced a resonant frequency shift with a maximal value of 1117 Hz for a single 100 nm NP, resulting in a deduced mass resolution of 740 ag (attograms = 10^‒18^ g) [72].

#### 3.2.2. Capacitive Sensors

Capacitive sensors are much more attractive for on-chip realizations due to their ease of fabrication and integration with other microelectronic and/or MEMS components and even in ultra-large scale integration (ULSI) ICs. However, the specific configuration and design of the electrodes is essential for optimal performance of such a particle sensor. A good analysis of such capacitive sensors for detection of large and medium size PM (1‒10 μm) was done by Carminati et al. [73]. Two main configurations are clearly possible: parallel and coplanar electrodes, the latter typically employing interdigitated electrodes (IDEs). In the first case the airborne particles will have to flow in-between the parallel plates, in which case the exact positioning of the particle in the channel does not matter due to the uniformity of the electric field between the parallel plates. In the second case, however, the electric field is highly nonuniform and in order to use it for particle sensing the particles must be either situated in the IDE’s immediate vicinity, or may even be deposited on top of them. Either way, the presence of a particle will result in a capacitance variation Δ*C* relative to the default case (without any particles), whose amplitude will depend on both the diameter *D* and dielectric constant ε_r_ of the particle. Therefore, the same variation ΔC could be ambiguously due to either a small particle with a large ε_r_, or, conversely, a larger particle but with a smaller ε_r_. Hence, performing a single Δ*C* measurement will not suffice, especially if both *D* and ε_r_ are desired to be deduced. Fortunately, this problem can be solved due to the fact that the dependency of Δ*C* on each variable has different variations: Δ*C* ∝ *D*^3^ but Δ*C* ∝ ε_r_. Thus, the problem can be solved by combining in cascade or in parallel multiple sensors of different geometries which provide more capacitive variations from which the necessary variables values can be extracted. However, this means that the particles will need to be sensed by all the structures making up the sensing architecture; settling/trapping on just a single sensor will not be desirable. It also means that the surface area ‒and thus the cost‒ of the sensor will also increase correspondingly, together with a necessary complication of the system’s functioning in order to allow such an operation mode [73].

The previously mentioned ambiguity may seem a major disadvantage when compared to the much more popular scattering-based particle detection. However, this is not the case, due to two factors. First, the same kind of ambiguity affects scattering detection as well due to the fact that the intensity of the scattered light also depends on both *D* and the particle’s refractive index *n* (which is intrinsically related to the dielectric constant as $n=\sqrt{\varepsilon_{r}}$). Second, the mathematics of light scattering is notoriously complex and difficult, and in order for the desired unknown parameter values to be extracted from scattering data many simplifying hypothesis also have to be applied, just as for capacitive sensing. For instance, the particles are assumed to be of spherical shape, of uniform composition, ideal dielectrics with no absorption of a certain refractive index value (or in a narrow range), and with no dispersion (i.e., no variation of optical properties as a function of wavelength). In practice, however, one or more of these assumptions do not correspond to the real characteristics of the particles. Consequently, the obtained results are only an estimation of the actual situation, with a limited precision. This probably explains why most of the portable particle detectors (which use exclusively scattering) do not indicate the precision of their measurement results.

The key drawback of the parallel plate geometry results from its intrinsic structure in which the electrodes are delimiting the flow channel between them. Hence, this structure can be used only for particles with diameters quite smaller than the distance between the plates in order to prevent clogging. This reduces the magnitude of the largest capacitance variation ΔC that can be sensed, which is paramount for detecting the smallest particles [73].

The planar structure with IDEs is more advantageous because it is easier to fabricate and allows the detection of particles with a wide range of sizes [73]. Although it no longer presents the risk of clogging, it can, however, be progressively dirtied by the gradual accumulation of particles over time, thus requiring periodic cleaning. Another disadvantage is the potential short-circuit between two (or more) electrodes caused by the deposition of conductive particles. One solution is to protect the IDEs with a very thin dielectric layer, i.e., with a thickness much smaller than that of the smallest particle to be detected. Another, more complex solution, is to use several detectors at the same time (just as a camera uses simultaneously multiple pixels), but this obviously imposes a much increased complexity for the system operation (as mentioned above) as well as for the subsequent read-out and signal processing circuits [13].

In order to understand better the performance of the planar capacitive structures, Carminati et al. performed COMSOL simulations [73] considering a large electrode width *W* = 100 μm, an inter-electrode gap *G* = 10 μm and the location of the particle to be detected of diameter *D* = 10 μm was fixed at a height *H* = 10 μm above the IDE. The results of their simulations are shown in Figure 8. It can be seen that Δ*C* decreased quasi-linearly as G increased, but increased in a logarithmic-like manner as the IDE’s electrodes length *L* increased, with little subsequent rise for lengths *L* > 30 μm. In contrast, as expected, Δ*C* increased very rapidly as the particle centroid-surface distance H decreased, due to the rapid decrease of the electric field’s strength away from the electrodes. Regarding the variation of Δ*C* with ε_r_, it can be considered quasi-linear for ε_r_ < 10 but quickly saturated nonlinearly after that, which the authors stated was due to the fact that extremely polarizable particles tend to behave like metallic ones. Measurements were subsequently performed using IDEs made of 100 nm/20 nm thick Au/Ti layers patterned on Pyrex wafers in order to test the possibility of capacitively detecting single microparticles in controlled and well-defined conditions. For this purpose, both static detection of two polystyrene beads each 20 μm in diameter and dynamic detection of talc micropowder particles were performed. A dedicated capacitance sensing circuit comprising a low-noise integrator front-end and synchronous lock-in detector combined with a differential configuration allowed achieving a suitable resolution of ∼1.2 aF (attoFarads = 10^‒18^ F) and the measured data matched the simulated values very well [73].

Capacitive sensors have the advantage that they can be easily integrated with circuits in CMOS ICs to provide a much more performant system in terms of sensitivity. Moreover, capacitive sensors can be used not just for PM detection but also for other applications, such as (bio)chemical, humidity and even temperature sensing. One such monolithic airborne PM detector for particles with a size down to 1 μm integrated both the interdigitated electrodes (500 μm long, realized in the fourth, last, metal level of a 0.35 μm CMOS fabrication process) and signal processing electronics on the same standard CMOS chip, enabling an improvement of the resolution by more than one order of magnitude [74,75]. It boasted a multichannel architecture comprising 32 sensors operating in parallel was used to reach a sensitive area of about 1.15 mm^2^ while achieving sub-aF resolution. A 1 μm value for both the width and the gap of the IDEs of each channel’s sensor had been optimized by FEA for the detection of 1 μm particles. The numerical simulations also estimated a variation Δ*C* = 0.7 aF for a particle with *D* = 1 μm and ε_r_ = 2, which increased linearly to more than 45 aF when the size of the particles increased up to 15 μm. This performance could be achieved by using two key elements. First, the large default capacitance of the IDEs (180 fF, due to the large length of the IDEs as well as of the parasitic coupling through the Si substrate) was canceled out by a differential structure. Second, a lock-in technique was employed, featuring a 6-bit digital-to-capacitance converter with a resolution of 150 aF for each arm in order to balance the differential input capacitance, as well as a DC bias network to reduce the low frequency noise. Consequently, this system could perform real-time detection, counting and sizing of PM with an equivalent diameter greater than 1 μm deposited on the interdigitated sensors, with resolution better than 100 zF (zeptoFarads = 10^‒21^ F) rms (average of 65 zF, with best obtained value of 45 zF), and a nonlinearity error of 15 aF. Practical measurements were performed using mineral talc characterized by a lognormal diameter distribution with a median of 8 μm and dielectric constant of 2.4. The talc particles were mechanically suspended in air above the chip at different concentrations and distances from the chip operated without package and Faraday cage. The results obtained after the deposition of single talc particles 2, 6 and 12 μm in diameter matched very well the simulated values and showed a linear dependence of the capacitance variation on the particle diameter in a log-log plot. Furthermore, by extrapolating the relation between ΔC and the diameter of particles and considering the resolution of 65 zF, the smallest detectable diameter of a talc particle with ε_r_ = 2.4 would be 0.3 μm (with a SNR = 1), which the authors stated that it is comparable with the detection limit of the state-of-the-art instrumentation based on laser scattering [74,75]. Further improvement, but for a single channel, was reported by using an integrated CMOS ultra-low-noise and wide-bandwidth (DC to 1 MHz) current amplifier that provided fA (femtoAmpère = 10^‒15^ A) current resolution and zF capacitance resolution which made possible nanoscale impedance measurements [76]. The chip, fabricated using 0.35 μm CMOS technology, exhibited a ~5 zF rms capacitive resolution and a 15 fA rms current resolution with a 1 kHz bandwidth. The minimum detectable signal was 40 zF (with a SNR = 10) which, given a maximum detection range of ±14 fF, results in a dynamic range of 129 dB and a corresponding sensitivity of 0.35 ppm. These excellent measured results confirmed that the chip would enable high-resolution capacitance detection for nanoscale measurements [76].

A different but related approach is to employ capacitance-to-frequency converters, a technique which can also be employed for any type of IDE-based capacitive sensors, not just PM detectors [77]. One such realization was a PM_2.5_ detector implemented using a 0.35 μm 2P4M CMOS technology with 3 V power supply. It boasted on-chip calibration that reduced the maximum linear error from 16.22% to 0.08%, and enabled measurements of particle concentrations in the range 6.64 to 55.62 μg/m^3^, for which the corresponding output frequency ranged between 0.66 to 3.98 MHz, with a sensitivity of 834.2 kHz/V [77].

Another report simulated the method for a 0.18 μm CMOS technology, but using a different approach [78]. The charge-based capacitance measurement employed a differential variation that was amplified using current mirrors, so that ultimately the capacitive sensor fed a current-controlled oscillator to convert Δ*C* into frequency-modulated pulses. However, instead of using the traditional charging of a capacitor, here the measured transient current was a periodic exponential waveform generated as the current-controlled oscillator quickly followed the transient current variations caused by small Δ*C* changes. Simulations showed a very linear dependence of the average number of pulses ‒up to ~500 pulses‒ as the sensing capacitance changed up to 16 fF during 10 periods for four different values of the reference capacitance (*C_R_* = 50, 100, 150 and 200 fF). The authors concluded that their chip offered a wide dynamic range of 16 fF and digital calibration which eliminated the need for the traditional analog switching [78].

#### 3.2.3. Impedance Sensors

Extremely few impedance-based particle sensors have been reported previously. Besides the DLD device of Huang et al. [40] (see again Section 2.3), only the previously mentioned work of Chiriacò et al. [60] also used impedance for particle sensing (see Section 3.2.1). However, neither was tested with airborne particles. The former was characterized with liquid suspensions of particles while the latter was tested with drops of particle suspensions that had been successively deposited and dried out onto the sensors surfaces. In contrast with the SAW sensors, the EIS sensors in the latter detectors were tested with drops containing different sizes of particles. For both 1 μm particles and 200 nm particles measured separately, the measured impedance values constantly increased almost linearly with subsequent depositions. When both 1 μm and 200 nm particles were mixed together, the impedance sensors output varied linearly with both the total mass and number of particles, with an estimated mass sensitivity of 45 Ω/ng and a LOD of ~2.8 ng. However, the impedance detection of 40 nm particles was not possible, and the authors hypothesized that this was due to particle desorption caused by a too small adhesion (roughly proportional to the particle radius) [60].

#### 3.2.4. Electrostatic Sensors

The electrostatic capture and detection of particles is one of the oldest methods which had been used extensively in industry, e.g., to remove soot and ash from exhaust fumes before they exit smokestacks. The same method was also employed to operate the Aerasense NanoTracer commercialized by Philips, considered to be a highly performant mature handheld instruments for personal monitoring of nanoparticles [13,79,80], and which is also mentioned in the last row of Table A1 in the Appendix A. Its structure and operating principle are illustrated in Figure 9. After the air is aspirated in the device by a small fan, the airborne particles are charged by a high-voltage stage that generates a strong electric field between a needle electrode and a surrounding screen. At the end, the charged particles are electrostatically attracted toward a grounded Faraday cage acting as a target connected to a low-noise current reader which will provide an output proportional to the charge of the arriving particles. Since the charge of a particle is proportional to its outer surface, the average current level thus measured will be indicative of the average particle size. The parallel-plate electrodes placed at the middle of the particles’ flow path allow the application of a transverse electric field, whose magnitude depends on the applied bias voltage *V*_pl_. When this transverse field is activated, some particles are deviated and no longer reach the electrometer. Under these circumstances a smaller current *I*_2_ is measured by the current reader than that obtained when *V*_pl_ = 0 (*I*_1_). Both current values are, however, important as the average number of particles is linearly proportional to the Δ*I* = *I*_1_–*I*_2_ difference, while the average detected diameter depends proportionally on the *I*_1_/(*I*_1_–*I*_2_) ratio. The Aerasense NanoTracer reportedly had a detectable particle size range between 20 to 120 nm and the minimum concentration of particles (set by the current resolution of 1 fA of the electrometer) was of about 1800 particles/cm^3^ with a response time of 10 s [13,79,80]. The sensitivity was *S_N_* = 900 particles/(cm^3.^fA) for particle concentration measurement, and *S_dp_* = 20.2 nm for particle size measurement, while the measurement resolution for particle size was 11 nm at *N* = 10,000 particles/cm*^3^*, *d_p,av_* = 50 nm, and Δ*I* = 1 fA [81].

An on-chip electrostatic-based particle detector targeted for monitoring submicron particles was realized using MEMS by Lim et al. [81]. The microsystem included a micro-VI designed to provide a cut-off diameter of 0.6 μm at an air flow rate of 0.33 L/min., followed by a micro-corona discharger stage (where a high voltage of at least 1.3 kV was applied between a bottom sharp tip and an upper planar electrode to charge the filtered airborne particles), and two collection electrodes that provided a current proportional to the concentration of the arriving charged particles. The entire particle detection chip was made of two epoxy-bonded wafers, a bottom Si wafer and a top glass wafer, respectively. The bottom Si wafer contained the sharp tip, its contacting electrodes and the 570 μm wide microfluidic channel patterned in SU-8 200 μm thick that was sealed by a top glass wafer with the sensing electrodes and inlets/outlets. The measured cut-off diameter was 50 nm smaller than the designed value, most probably due either to the deviation of major to minor flow ratio from the 90:10 value or to slightly different channel dimensions realized in the practical device. The measurement sensitivity is not specifically given, but from the data given in the paper a value of ~9×10^‒6^ pA per particle/cm^3^ can be deduced [81]. A subsequent chip realized by the same group had the same structure, but the micro-VI was designed in two versions, namely with a cut-off diameter of either 600 nm or 1 μm [82]. Tests were then performed with DOS particles with sizes ranging from 100 to 600 nm at 3 different concentrations at input air flow rates of 0.3 L/min or less. Measurements indicated practical cut-off diameters of 0.55 μm and 1.1 μm, respectively, and when the corona discharger was biased with a voltage of 1.3 kV, the current due to charged particles measured with an electrometer at the sensing electrodes was linearly proportional to the number concentration (N/cm^3^) of DOS particles 500 nm in diameter that had been injected at a flow rate of 0.3 L/min. The sensitivity calculated from measured data was 8 × 10^‒7^ pA per particle/cm^3^ and the collection efficiency was almost 100% when a cut-off diameter of 1 μm was used for collecting 100 nm particles. The authors concluded that their device had potentially similar performance in terms of collection efficiency when compared with existing instruments but at a much more advantageous/reduced cost and size/volume due to its inherent compact miniaturized structure/design. A major drawback, caused by the same reduced scale, was a maximum sampling rate 10 to 100 times larger than that of similar existing instruments, which would increase too much the power consumption of a sensor supposedly suitable for portable applications. Also, it was very challenging to accurately measure the extremely small current proportional to the number concentration of the detected particles, as it was close to the background noise and thus prone to relatively large fluctuations, requiring a very careful measurement set-up and adequate shielding [82].

Another chip employed only the Si micromachined tip with its top plate as well, in order to realize an unipolar charger targeted to measure submicron aerosol particles [83]. The fluidic channel was realized in a SU-8 layer 200 μm thick which also served as spacer between the top glass wafer and the bottom Si wafer. Unlike the previous two realizations, this chip did not include any IF or VI structure for preliminary size-based particle separation. After an initial check of the current‒voltage (*I*-*V*) characteristic, the particle loss and charging characteristics were assessed with NaCl particles of sizes in the range 30‒130 nm at an air flow of 0.3 L/min. The voltage applied to the corona tip was between 1–4 kV, but the optimal operating range was between 2‒3.5 kV, and above 3.8 kV a spark-over phenomenon manifested. The particle loss was 2.1% for 20 nm particles but rapidly increased up to 16.6% for the smallest measured particle size of 30 nm due to diffusional losses as the particle size decreased. Both the calculated and simulated data for the corona currents matched the measured results well. However, the geometric mean diameters calculated from measured data were 12% larger than the results provided by a reference scanning mobility particle sizer (SMPS) and the deviations were due to the fluctuations and resolution limit in the read-out of the collected current data, which may be improved if more sensitive current measurement tools and/or a larger flow rate could be employed. The experimental values of the *p*^.^*n_p_* product were found to be proportional to the square of the particle diameter (*p* is particle penetration, with *p* = 1‒*L*, i.e., it is the complementary of particle loss *L*, and *n_p_* is the particle charge number due to field charging). The authors concluded that the measured data of particle loss, average particle charge, number concentration, and geometric mean diameter were in good agreement with the simulated results and proved the ability of the device to provide particle size distributions [83].

## 4. Conclusions

Airborne particulate matter presents a significant health risk, and the importance of air quality assessment has increased rapidly in the last decades due to the increased pollution, urbanization, and ecological disasters such as forest wildfires and/or the ‒many times‒ uncontrolled land clearings, like those that occurred in South-East Asia in 1997, 2013 and 2015, or the recent ones that occurred on large areas since 2017 in U.S.A. (California), Australia, Greece, Sweden, Portugal and in other countries [84,85,86,87,88,89,90,91,92,93,94,95,96,97]. Therefore, the need for portable PM monitors has increased constantly, not just for pollution monitoring but also for tracking personal PM exposure, and miniaturized/on-chip PM separators and detectors are paramount for the realization of such compact yet efficient systems.

Numerous such down-scaled devices have been realized, particularly in the last years, based on a variety of principles for particle separation and detection, each with their own advantages and weaknesses. In the great majority of cases, particle separation is first performed in order to eliminate particles with mean aerodynamic diameters larger than the desired cut-off (2.5 μm or 10 μm for PM_2.5_ or PM_10_, respectively). This is typically done using just a single micro-VI or IF, but cascaded stages can also be employed. Lately, particle separation based on aerodynamic principles and deterministic lateral displacement filtering were also demonstrated, although their performance characteristics ‒especially particle separation efficiency‒ may not be as good as those of the more traditional VI or IF realizations. Cyclone separators can also be employed, but due to the intrinsic nature of their design they are very difficult to be miniaturized, as they require large volumes in which the desired airflows can be achieved, as well as a powerful enough pump or motor which can set the air in motion.

Interestingly, two key components of any fluidic system ‒pump and valves‒ reportedly have not yet been integrated with and applied in on-chip airborne particle monitors, although there is a large literature of such MEMS-based devices suitable for this purpose.

The particle detectors can be optical or electric. The former type of devices relies exclusively on scattering, but this method requires not only an extremely sensitive detector but also a very complicated mathematical apparatus to extract the desired data from measurements. Because many simplifying assumptions have to be considered in order to greatly simplify this task, many times the obtained result is an estimation with a relatively limited precision, e.g., ~10‒15%, which can be improved only if significant complication of the measurement set-up and/or more complicated and more accurate algorithms can be used. Furthermore, adopting a scattering-based detector in a new product intended for commercialization may be very challenging in terms of the potential originality and innovation it can offer, since the great popularity and exclusive usage of scattering-based detectors in almost all hand-held devices available now on the market means that a very large number of existing patents are already covering this field.

Electrical particle detectors have a long history. On one hand, SAW- and QCM-based devices have been used and successfully demonstrated for a long time. However, they also have some important limitations. Firstly, they are very difficult to be integrated in (predominantly) monolithic structures due to the need of a piezoelectric substrate which prevents the fabrication of electronic components and severely impedes the realization of other micromachined structures in it. Secondly, and maybe more importantly, the more recent reported realizations employing MEMS/NEMS-based resonant structures (such as cantilevers or even nanorods) can offer not only a hugely improved sensitivity with two or three orders of magnitude but can also be easily integrated with signal processing circuits or other micromachined components. However, the latter typically have extremely low particle capture efficiencies, which in principle could be improved using large arrays of many identical elements, but this would also require a much more complicated circuitry for proper array scanning and for the subsequent signal processing.

On the other hand, capacitive detectors, particularly the IDEs, are easy to fabricate and integrate together with complex circuits for signal processing and data conditioning, and some very sensitive solutions have been reported recently. Furthermore, such solutions can be employed not only for particle sensors but for any sensors based on IDEs. However, capacitive particle sensors have a few disadvantages. First, they can be used only in a relatively narrow range of sizes (dictated by the smallest feature size of the IDEs that can be patterned photolithographically). Second, the particles must be brought very close to the detector surface, and in most cases they are trapped onto a sticky layer that must be deposited *a priori* on the surface. This, in turn, leads in time to gradual accumulation of material onto the detector surface, demanding also the inclusion of an auto-zero circuit in the signal processing part so that it should self-adjust its gain in order to compensate for the correspondingly reduced sensitivity. It is true, though, that the material accumulation on the detector’s surface is a general problem shared by almost all particle detectors and which is extremely difficult to overcome. As was shown earlier, cleaning by PDMS casting has been successfully demonstrated in one case [71], although this solution may not be easily adopted for mass produced devices intended for usage by the general public. Another impediment could be the need to use two or more detectors in order to eliminate the size versus material properties ambiguity of the measurement, and the corresponding complications it entails for the system operation and for the subsequent signal processing. Lastly, but not the least, the larger the targeted sensitivity and resolution the greater the importance of the parasitic components, whose elimination (or at least minimization) is far from trivial, particularly when the sensor is integrated with an ULSI IC. Nevertheless, very sensitive capacitive detectors integrated with CMOS ICs have been demonstrated.

For nanoparticles, electrostatic separators have been widely employed industrially and are well known and even handheld instruments are now available commercially. Unfortunately, their inherently complicated design and the need for very high voltages (which still may remain on the order of kV even for down-scaled devices) makes them much less feasible or attractive for further down-scaling, let alone for on-chip integration. Still, a few such miniaturized on-chip realizations have been reported, but they have inherent drawbacks as well. Exactly because of their very small size they require special solutions for the extremely small current/charge measurement that is now close to noise level, while the measurement/sampling time has to be one or even two orders of magnitude longer.

We can conclude that the miniaturized and on-chip particle detectors are not just a fascinating and a much less known field (compared to the significantly larger number of reports and devices in other areas of microfluidics/micromachining/MEMS/miniaturized systems), but one whose importance is constantly increasing. This is because air quality assessment and airborne particle detection has experienced significantly increased dynamism in the last years, and it is reasonable to expect that the trend will continue. Hence, we can anticipate that new and more performant devices may appear in the near future, some probably based on new principles, as the demand for such systems, especially of miniaturized PM detectors for personal use or in ‘smart homes’, may correspondingly increase substantially. 

## Figures and Tables

**Figure 1 micromachines-10-00483-f001:**
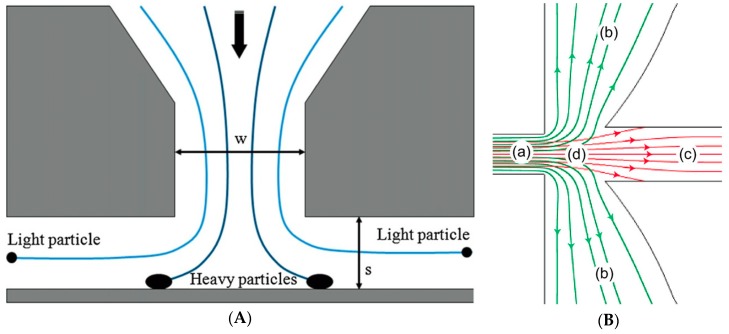
Cross-sectional views of: (**A**) The typical ‘Marple’-type impactor filter (IF): due to their (much) larger inertia, heavy and/or large particles cannot follow the sudden abrupt change in airflow direction and are collected on the impaction plate which may be coated with a sticky layer for optimal results [14] © IOP Publishing. Reproduced with permission. All rights reserved. (**B**) Structure of a microfabricated virtual impactor (VI) realized by Paprotny et al. and simulated flow lines in it [21]. The explanation of the labels and colors is given in the text. Reprinted from [21] Copyright (2019), with permission from Elsevier.

**Figure 2 micromachines-10-00483-f002:**
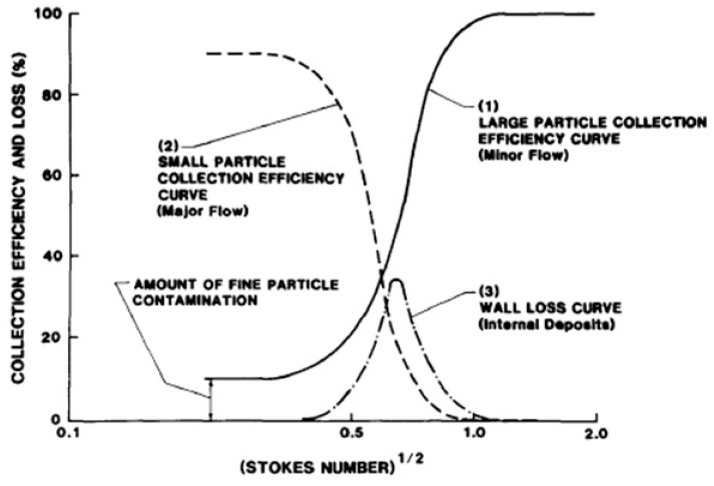
Performance characteristic curves for an IF or a VI [17]. Reprinted from [17]. Copyright (2019), with permission from Elsevier.

**Figure 3 micromachines-10-00483-f003:**
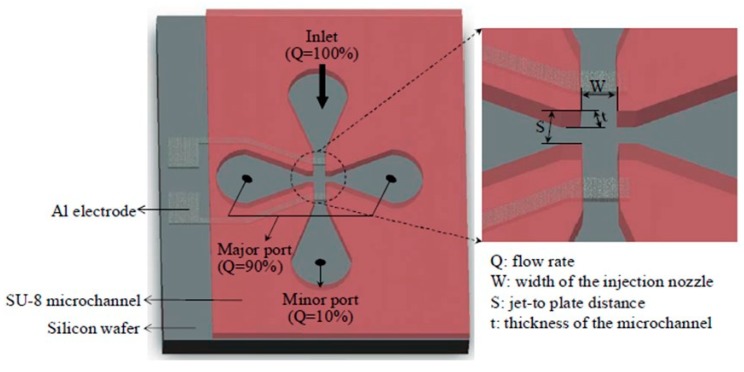
The electrically tunable VI: the entire microfluidic chip (left) and zoom-in detail of the tuning element at the narrow throat of the VI (right). © 2019 IEEE. Reprinted, with permission, from [27].

**Figure 4 micromachines-10-00483-f004:**
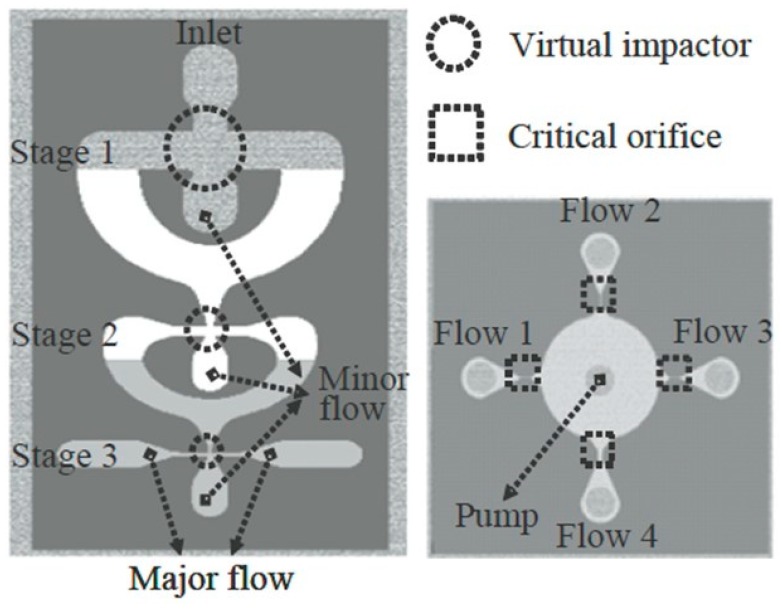
Three-stage cascade VI (left) and flow rate distributor (right) microfabricated in stacked SU-8 layers; © 2019 IEEE. Reprinted, with permission, from [32].

**Figure 5 micromachines-10-00483-f005:**
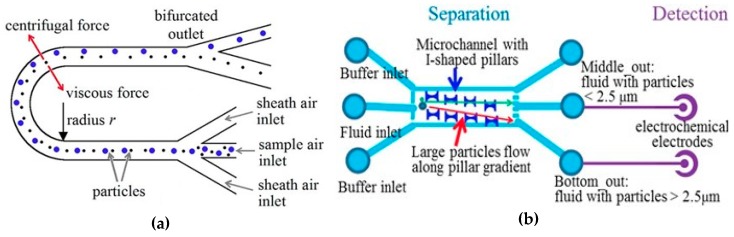
(**a**) The structure and operation principle of the centrifugal force-based aerosol separation system; © 2019 IEEE. Reprinted, with permission, from [37]; (**b**) Structure and operating principle of the microfluidic chip employing deterministic lateral displacement (DLD) of particles; © 2019 IEEE. Reprinted, with permission, from [38].

**Figure 6 micromachines-10-00483-f006:**
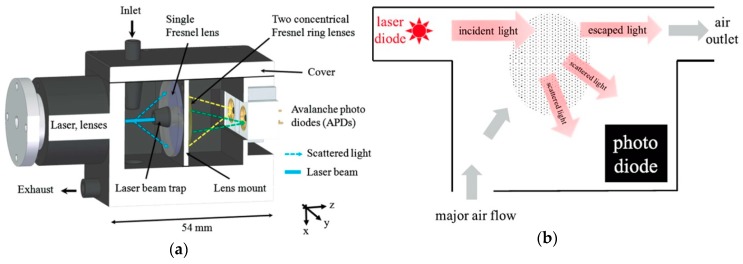
Scattering-based systems for PM detection: (**a**) The miniaturized setup with three Fresnel lenses of Schrobenhauser et al. [14,43] © IOP Publishing. Reproduced from [14] with permission. All rights reserved. (**b**) Schematic structure of the chip-based realization of M. Dong et al.; © 2019 IEEE. Reprinted, with permission, from [47].

**Figure 7 micromachines-10-00483-f007:**
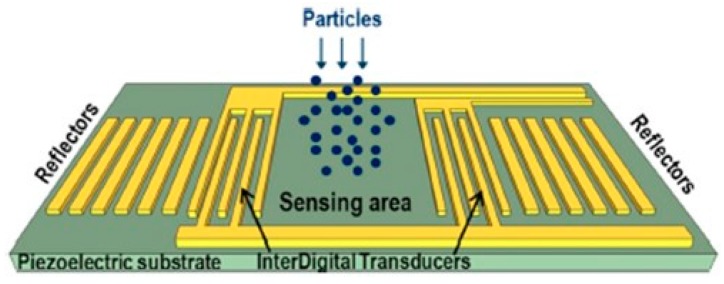
Structure of a surface acoustic wave (SAW) resonator-based particle detector [48]. Reprinted from [48]. Copyright (2019), with permission from Elsevier.

**Figure 8 micromachines-10-00483-f008:**
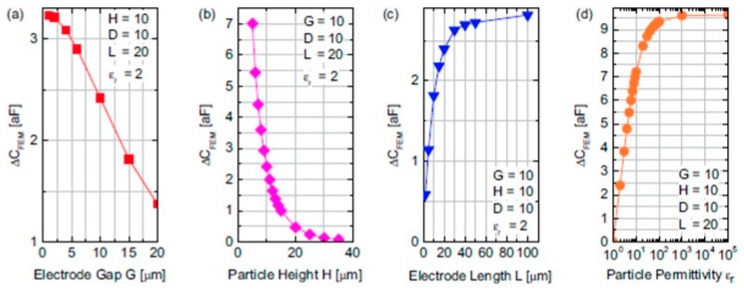
Simulation results showing the dependence of the capacitance variation signal ΔC (in [aF]) of an IDE sensor on: (**a**) the gap G between the electrodes; (**b**) the vertical distance H of the centroid of a PM_10_ particle from the bottom electrode level; (**c**) on the electrode longitudinal length L, and (**d**) on the particle relative dielectric constant ε_r_ [73]. Reprinted from [73] Copyright (2019), with permission from Elsevier.

**Figure 9 micromachines-10-00483-f009:**
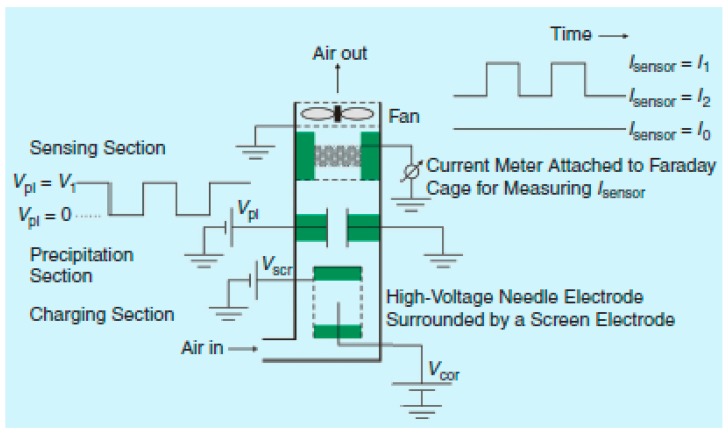
Schematic representation of the Philips Aerasense Nanotracer electrostatic handheld monitor of airborne nanoparticles; © 2019 IEEE. Reprinted, with permission, from [79].

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
