# Peer review of "Microfluidic and Micromachined/MEMS Devices for Separation, Discrimination and Detection of Airborne Particles for Pollution Monitoring"

_micromachines, 2019, doi:10.3390/mi10070483_

Round 1
Reviewer 1 Report
This paper reviews the research on microfluidic chips related to the separation, identification and detection of airborne particles. The author did a timely job and this review article is a good inspiration for scientists working on microfluidic airborne particle analysis. However, as mentioned in the title, the author wants to summarize "airborne particles" and a large class of airborne particles, which are bioaerosols such as airborne bacteria, viruses, fungi and spores. Several publications report microfluidic-based bioaerosol analysis. Here is a list of part of these works.
1. Microfluidic system for rapid detection of airborne pathogenic fungal spores. ACS Sensors. 2018.
2. A sample-to-answer labdisc platform integrated novel membrane-resistance valves for detection of highly pathogenic avian influenza viruses. Sensors and Actuators B-Chemical. 2018.
3. A novel microfluidic module for rapid detection of airborne and waterborne pathogens. Sensors and Actuators B-Chemical. 2018.
4. A semi-quantitative method for point-of-care assessments of specific pathogenic bioaerosols using a portable microfluidics-based device. Journal of aerosol science. 2018.
5. Detection of pathogenic microorganisms by microfluidics based analytical methods. Analytical chemistry. 2018.
6. Highly enriched, controllable, continuous aerosol sampling using inertial microfluidics and its application to real-time detection of airborne bacteria. ACS Sensors. 2017.
7. Droplet-based microfluidics detector for bioaerosol detection. Aerosol science and technology. 2017.
8. Rapid capture and analysis of airborne staphylococcus aureus in the hospital using a microfluidic chip. Micromachines. 2016.
9. High-throughput microfluidic device for LAMP analysis of airborne bacteria. ACS sensors. 2016.
10. Continuous aerosol size separator using inertial microfluidics and its application to airborne bacteria and viruses. Lab on a chip. 2015.
Therefore, I suggest that the author should add at least one or a few paragraphs to discuss bioaerosols, as this article focuses on the analysis of airborne particles in microfluidic platforms. After major revisions, this article can be considered for publication.
Author Response
Thank you very much for your comments, please kindly see my responses in attached file

Reviewer 2 Report
This review should be re-considered for publication after a major revision. A number of major corrections should be considered.
First and foremost, the way it is currently written, the citations are not nearly clear enough for a review paper. Through much of the paper, entire paragraphs are written with the citation only appearing at the end, while important information that should be cited has obviously come earlier in the paragraph. Also, some citations are just “author name et al.” with no link to the citation number - with numerical citations throughout the paper, these bare author names are impossible to find in the bibliography, especially with no date cited either. In a review, citations should be comprehensive, clear to the ready, and accurate.
There should be a lot more figures and schematics from the papers you cite. You describe many devices in words only, and it is very hard to follow along and visualize these devices.
For the figures you do have, do you need to explicitly state that they are used with permission of the respective publishers?
Make sure the figures are of sufficient resolution - Figures 3 and 6 look a little pixelated to me.
The abstract lays out a good goal: to review micro devices that are used for PM/air quality monitoring. But then the rest of the paper doesn’t seem to connect any of the devices you talk about to this specific application. This area is where you need to add a lot of your own analysis. For example, compare the micro devices you review to existing PM/air quality monitoring devices. A table of sample flow rates, lower detection limit, size range of detection, cut sizes, precision and accuracy, etc. would be a great addition.
To reiterate the previous point, several of the devices you review are not given with sufficient detail that an aerosol scientist would know what those devices are good for. Presumably, all of the devices you discuss are used to measure PM as way to determine air quality. However, you are not explicit about the purpose(s) of each device. With the different levels of performance, you should be able to assess how well each device should perform in the real world too.
Another observation about the devices you review - you give a varying amount of detail about the devices and its not clear why. There needs to be a unifying purpose to discussing each one, which would be settled if you address the previous two items.
There are many numbers and units that you quote that might not be known to all your audience. For example, I have no idea what Hz/ng or Hz/microgram really mean, what values are desirable. How do these units relate to actual, real-word PM monitoring devices? When quoting values like these, make sure to explain the importance of the numbers.
Some more minor, yet equally important, changes that should be considered:
Do not use “this” as a noun in a sentence as it add ambiguity.
There are several uses of “since” when “because” is the proper word to use.
There are a number of 1-sentence paragraphs throughout the paper, which is generally considered poor grammar.
You use the word “neglected”, and yet talk about quite a few devices and have a decent number of references. Your conclusion even uses the word “numerous” on line 753. Just because a research field is small or new and up-and-coming doesn’t mean that it is neglected.
Your title just says “microfluidics” and yet you review a number of devices that would be considered “miniature systems” or “micromachines” or “MEMS” - consider a more broad title that captures all of what you review.
Specific comments:
Line 31 - missing subscripts in the formatting
Line 48 - spell out “%”
Line 51 - “foetal” is misspelled?
Line 76 - spell out “&”
Line 104 - While not technically wrong (I don’t think), the use of “inert” here might be confusing to some readers.
Line 110 - typo in citation numbers
Line 129-131 - confusing - were the particles sampled 10 microns or not?
Line 159 - which “authors”?
Line 207 - what is “SOI”
Line 222 - too many “desired”s
Line 238 - The safety and cleanliness concerns are not “obvious” to me.
Line 247 - what is “FEA”?
Line 250 - “reported previously” - where, when, and by whom?
Line 265 - use of “radially” is awkward
Line 277 - use of “flown” is awkward
Line 296 - comma not needed
Line 336 - “This enabled to measure” is not right
Line 359 and elsewhere - “30…250” should be “30-250”, a more clear way to present a range of values
Line 359 - clarify, counting efficiency?
Line 427 - what is “AT”?
Line 429 - what is “high sensitivity”?
Line 431 - what is “ST-cut”?
Line 449 - “CMOS” is never defined; it might be common enough that it doesn’t need to be, but consider it to be complete
Line 473 - what does “ST-X” mean?
Line 519 - Is a sampling efficiency of only 1.33% correct? Isn’t that incredibly small? Wouldn’t that mean this is a pretty useless device?
Line 529 - quantify “very good correlation”
Line 569 - what is “ag”?
Line 593 - “much popular” is awkward
Line 613 - too many “progressive”s
Line 614 - “requiring for” is awkward
Line 620 - I can’t visualize what this paragraph is talking about. A schematic of the device and/or a plot of the modeling results with the important conclusions annotated on the figure would help.
Line 647 - what is “zF”?
Line 658-661 - definitely need to describe the units here and how these are “excellent”
Line 684 - should be “operating principle”
Line 702 - spell out “submicron”
Line 711 - “2 years later” is an unnecessary detail
Line 724 - why is the larger max sampling rate a drawback?
Line 733 - what is the “I-V characteristic”?
Line 739 - SMPS is not defined
Line 745 - I’m confused - simulations proved its a usable device but the actual comparison to SMPS did not?
Line 749 - are land clearings really “uncontrolled”?
Line 809 - should be “orders”
Author Response
Thank you very much for your comments, please see my responses in attached file.

Round 2
Reviewer 1 Report
After revision, this article has now met the published requirements and is recommended for publication.